# Identity-Preserving Image-to-Video Generation via Reward-Guided Optimization

## Abstract

Recent advances in image-to-video (I2V) generation have achieved remarkable progress in synthesizing high-quality, temporally coherent videos from static images. Among all the applications of I2V, human-centric video generation includes a large portion. However, existing I2V models encounter difficulties in maintaining identity consistency between the input human image and the generated video, especially when the person in the video exhibits significant expression changes and movements. This issue becomes critical when the human face occupies merely a small fraction of the image. Since humans are highly sensitive to identity variations, this poses a critical yet under-explored challenge in I2V generation. In this paper, we propose Identity-Preserving Reward-guided Optimization (IPRO), a novel video diffusion framework based on reinforcement learning to enhance identity preservation. Instead of introducing auxiliary modules or altering model architectures, our approach introduces a direct and effective tuning algorithm that optimizes diffusion models using a face identity scorer. To improve performance and accelerate convergence, our method backpropagates the reward signal through the last steps of the sampling chain, enabling richer gradient feedback. We also propose a novel facial scoring mechanism that treats faces in ground-truth videos as facial feature pools, providing multi-angle facial information to enhance generalization. A KL-divergence regularization is further incorporated to stabilize training and prevent overfitting to the reward signal. Extensive experiments on Wan 2.2 I2V model and our in-house I2V model demonstrate the effectiveness of our method. Code will be released to advance the related research.

## 1 Introduction

Identity preservation is a widely popular and explored topic in text-to-video (T2V) generation. Numerous recent methods (He et al., 2024; Liu et al., 2025c; Yuan et al., 2025; Wang et al., 2025; Li et al., 2024; 2025; Song et al., 2025; Xie et al., 2025; Zhang et al., 2025; Zhong et al., 2025; Jiang et al., 2025) attempt to generate high-fidelity videos that maintain consistent identity from user-provided reference subjects, often by incorporating auxiliary identity modules and carefully curated datasets. However, it remains a persistent yet underexplored challenge in image-to-video (I2V) generation. Despite diffusion transformer-based models such as CogVideoX (Yang et al., 2024), HunyuanVideo (Kong et al., 2024), and Wan (Wan et al., 2025) have significantly improved image-to-video (I2V) generation quality, they still struggle to faithfully preserve the identity of the input image, especially human faces. A significant challenge arises when the reference image contains a low-resolution human face and the motion prompt involves large-scale movements (e.g., jumping, turning). Under these conditions, as illustrated in Fig. 1, the propagation of errors across frames results in progressive identity degradation, causing the generated subject to drift from the appearance in the initial frame.

One might consider explicitly injecting identity features into the model—a practice commonly used in subject-driven generation. However, since the identity is already fully encoded in the first frame, the challenge lies not in the absence of information but in its preservation. Merely adding more identity cues is a redundant exercise that does not tackle the root cause of temporal degradation. Moreover, these supervised finetuning (SFT) methods suffer from exposure bias, a discrepancy between training and inference dynamics caused by training on ground-truth inputs but self-generated inputs at inference, leading to error accumulation and identity drift. In addition, such architecturally-

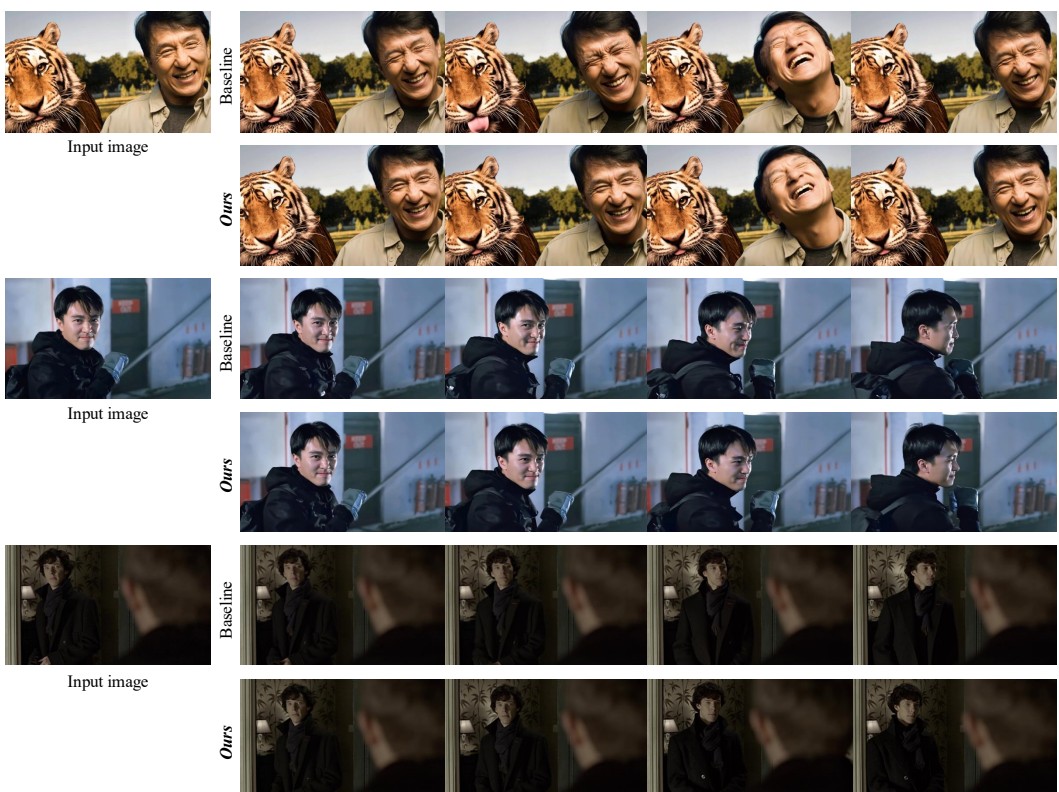

Figure 1: Given a input image with human faces, our method can achieve superior identity consistency in generated. **(Zoom in for best view)**

intrusive methods are inherently single-subject by design and struggle to scale to scenes with multiple individuals, making them impractical for real-world applications. This raises a natural question: Can we enhance the identity preservation capabilities of a general-purpose foundation I2V model without altering its architecture or compromising its original abilities?

In this paper, we present an image-to-video generation framework designed to maintain identity fidelity without extra identity networks or facial attribute injection but from the perspective of reinforcement learning. Prior works such as ReFL (Xu et al., 2023), DRaFT (Clark et al., 2023), AlignProp (Prabhudesai et al., 2023) and VADER (Prabhudesai et al., 2024) directly supervise the final output of a diffusion model using differentiable reward functions. These reward-guided approaches enable effective adaptation of diffusion models to generate videos that better align with task-specific objectives, thereby achieving better alignment with desired outcomes. Our approach performs direct reward optimization by leveraging a face identity model (Deng et al., 2019) as the reward model. To eliminate exposure bias, we thus train on-policy by initializing trajectories from pure Gaussian noise and backpropagates the identity reward through the last denoising steps of the sampling chain. This trajectory-aware optimization yields robust identity improvements, faster convergence, and better stability. To prevent reward hacking, we propose a robust multi-view facial scoring mechanism that leverages facial feature pools constructed from videos, substantially improving generalization to pose, expression, and lighting variations while mitigating the copy-paste phenomenon. We also incorporate a KL-divergence regularization that constrains the deviation of the tuned model from the original diffusion model at each gradient step. This multi-step regularization better preserves the model's original capabilities while enhancing identity consistency.

We evaluate our methodology on both Wan 2.2 I2V model (Wan et al., 2025) and our in-house 15B I2V model using comprehensive quality and identity preservation metrics, comparing against state-of-the-art methods for identity-preserving video generation. Extensive experiments demonstrate that our approach generates high-quality videos with significantly improved identity consistency.

## 2 RELATED WORK

**Identity preservation in video generation.** Current identity-preserving video generation methods mainly focus on T2V generation, relying on additional modules to inject identity information. ID-Animator (He et al., 2024) and ConsisID (Yuan et al., 2025) encode ID-relevant information with a face adapter. FantasyID (Zhang et al., 2025) introduces 3D facial geometry to preserve structure. Concat-ID (Zhong et al., 2025) and Stand-In (Xue et al., 2025a) concatenate reference face tokens at the end of video token and employ 3D self-attention to fuse them. Some methods (Wang et al., 2025; Xie et al., 2025; Song et al., 2025) propose mixture of facial experts to dynamically fuse identity, semantic, and detail via a adaptive gate. PersonalVideo (Li et al., 2024) utilizes per-scene LoRA adapters for video personalization, but it requires finetuning the pretrained model for each new person during inference. Unlike T2V generation, I2V models naturally support first-frame conditioning for better controllability. Adding extra modules to improve identity preservation would disrupt the original structure of foundation models and increase complexity. To address this, our method directly learns feedback from an identity reward function, producing videos that more faithfully meet identity-specific goals and maintain identity consistency.

**Reinforcement learning for diffusion models.** To tailor models to user needs, models are often fine-tuned using carefully curated datasets or perform reinforcement learning (Prabhudesai et al., 2024; Liu et al., 2025a;b; Xue et al., 2025b). Compared to supervised finetuning, which relies on the standard diffusion training loss, reinforcement learning offers a more direct and efficient optimization pathway by leveraging a reward model or curated preference data. While the former approach applies a general, pixel-level reconstruction signal, RL provides targeted feedback that directly addresses high-level perceptual attributes, such as identity consistency. This allows for lightweight, precise adjustments to the model's output distribution, correcting specific failure modes without extensive retraining. Direct preference optimization (DPO) (Rafailov et al., 2023; Wallace et al., 2024) skips a reward model and learns directly from human preferences, simplifying the pipeline but only optimizes relative preference ranking, with gradients derived from "positive negative" differences and susceptible to confounding factors such as aesthetics and clarity, lacking absolute calibration. MagicID (Li et al., 2025) leverages DPO to improve identity consistency for T2V generation, but still requires per-scene LoRA adaptation and per-person fine-tuning of the pretrained model, which substantially limits its practicality. While Group-based Reward Policy Optimization (GRPO) (Shao et al., 2024) stabilizes RL via intra-group advantage normalization, its efficiency in I2V is limited by low response diversity within a single prompt, resulting in highly similar video samples that undermine the utility of group-wise advantage estimation. For our goal of enforcing facial identity consistency, a metric that can be reliably quantified by a reward model, a more direct reward feedback learning paradigm is both better suited and more efficient. Therefore, we introduce the first facial reward feedback framework for I2V that directly optimizes gradients for identity preservation.

## 3 PRELIMINARY

**Diffusion Models.** Diffusion models are a class of generative models that learn to generate data by gradually denoising latents that are initialized from pure noise. The process consists of two phases: a forward diffusion process and a reverse denoising process. In the forward process, the data $x_0$ is gradually corrupted by adding Gaussian noise over $T$ time steps to obtain $x_T \sim \mathcal{N}(\mathbf{0}, \mathbf{I})$. This process is modeled as a Markov chain with fixed variance schedule $\beta_t$:

$$q(x_t|x_{t-1}) = \mathcal{N}(x_t; \sqrt{1-\beta_t}x_{t-1}, \beta_t\mathbf{I}), t = 1, ..., T. \tag{1}$$

The reverse process is learned by a neural network that approximates the reverse conditional distributions $p_\theta(x_{t-1}|x_t)$. Diffusion models learn the denoising network $\epsilon_\theta$ by minimizing the following re-weighted variational lower bound of the marginal likelihood (Ho et al., 2020):

$$\mathcal{L}_\theta = \mathbb{E}_{t,x_0,\epsilon}[||\epsilon - \epsilon_\theta(x_t, t)||^2], \tag{2}$$

where $x_t = \sqrt{\bar{\alpha}_t}x_0 + \sqrt{1-\bar{\alpha}_t}\epsilon$, $\epsilon \sim \mathcal{N}(\mathbf{0}, \mathbf{I})$, $\alpha_t = 1 - \beta_t$, and $\bar{\alpha}_t = \prod_{s=1}^t \alpha_s$. Once trained, samples are generated by reversing the diffusion process using the learned network $\epsilon_\theta$.

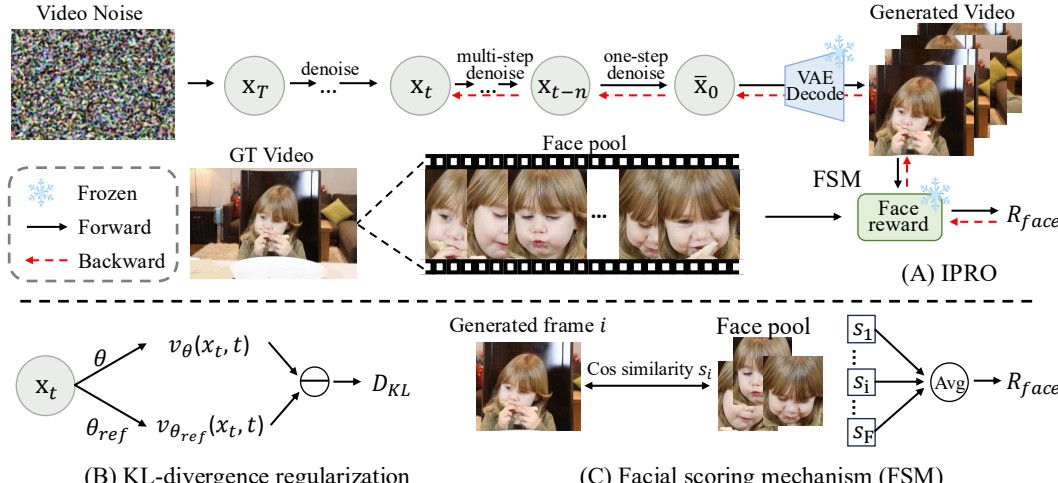

Figure 2: **Overview of our method.** (A) IPROpredicts $\bar{x}_0$ from the noise input $x_T$, and the prediction is visualized through a frozen VAE decoder and scored by a face reward model with our facial scoring mechanism (C). This reward signal is used to update the trainable parts of the model, thereby steering the generation process to produce videos with consistent identity. (B) We further incorporate a KL-divergence regularization to alleviate reward hacking.

## 4 METHODOLOGY

This work introduces a novel video diffusion model based on reinforcement learning to enhance identity preservation in I2V generation. The overall pipeline is illustrated in Fig. 2. Section 4.1 details reward-model strategies for maintaining facial identity consistency. Section 4.2 then introduces a facial scoring mechanism which treats faces in ground-truth videos as multi-view facial feature pools, thereby enhancing generalization and curbing copy-paste phenomenon. Finally, Section 4.3 incorporates a KL-divergence regularization to stabilize training and prevent overfitting to the reward signal.

### 4.1 IDENTITY-PRESERVING REWARD.

**Facial reward feedback learning.** We propose a direct and effective approach for adapting video diffusion models preserve facial identity throughout generated video sequences, by leveraging a differentiable facial reward function. The core idea is to guide the generation process through an explicit optimization objective that maximizes face identity consistency between the generated video frames and ground-truth video frames. To achieve this, we compute a facial reward score $R_{face}$ based on the cosine similarity between the Arcface (Deng et al., 2019) embeddings of the GT video and the generated video. ArcFace (Deng et al., 2019) has been widely recognized for its strong discriminative power in capturing fine-grained facial features, making it well-suited as a perceptual metric for identity preservation. Our goal is to fine-tune the parameters $\theta$ of a diffusion model such that videos generated by the sampling process maximize the differentiable reward function $R_{face}$:

$$J(\theta) = \mathbb{E}_{x_T \sim N(\mathbf{0},\mathbf{I})}[R_{face}(sample(\theta, x_T))] \tag{3}$$

where $sample(\theta, x_T)$ denotes the sampling process from time $t = T \to 0$.

Prior identity-preserving approaches typically rely on supervised fine-tuning with teacher forcing, which induces exposure bias: during training the model conditions on intermediate states derived from real samples, whereas at inference it must condition on its own generated states. This will be magnified as a long-term consistency issue in video identity preservation: small gradual errors cannot be seen and corrected by the training process, ultimately manifested as identity features drifting over time, being attracted by the "average face", and identity jumps occurring under occlusion/extreme postures. By using facial reward feedback learning, starting from random noise and generating and learning in an inference manner, the training distribution can be aligned with

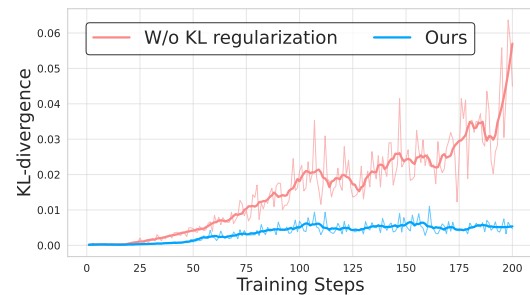

Figure 3: **The effect of KL regularization on KL divergence across training steps.** The model trained with KL regularization (blue) maintains a low and stable divergence, whereas the model without regularization (red) exhibits a rapid and volatile increase.

the inference distribution, directly optimizing the long-term identity consistency objective, thereby significantly alleviating the damage of exposure bias to identity preservation.

We efficiently learn the facial reward by backpropagating its gradient to update the diffusion model parameters $\theta$. To reduce memory and accelerate optimization, we adopt the DRaFT (Clark et al., 2023) truncation strategy, backpropagating only through the last $K$ sampling steps. Since identity preservation is driven by appearance details, and AlignProp (Prabhudesai et al., 2023) shows that later timesteps are most influential for these fine-grained details, this truncation does not degrade performance. This concentrates capacity where identity cues reside, reduces memory/compute, and prevents drift of global dynamics, and truncated gradient is given by:

$$\nabla_\theta R_{face}^K = \sum_{t=0}^K \frac{\partial R_{face}}{\partial x_t} \cdot \frac{\partial x_t}{\partial \theta} \tag{4}$$

### 4.2 FACIAL SCORING MECHANISM

We introduce a facial scoring mechanism (FSM) that treats the faces in the ground-truth video as a reference pool. Previous methods (Li et al., 2024; Xie et al., 2025; Yuan et al., 2025) computes identity loss by measuring face similarity either between generated frames and a reference image or between generated frames and their time-aligned ground-truth frames. However, the former encourages copy-paste phenomenon (the human in the generated video strictly maintain the expression in the given first frame) and suppresses facial expression diversity, while the latter provides weak supervision under teacher forcing, since the generated frame is already close to its corresponding ground-truth frame. In contrast, thanks to generating videos from random noise, we can treat the faces of all frames in the ground-truth video as a pool and, for each generated frame, calculate the average face similarity to all ground-truth frames. This objective encourages the people in the generated video to resemble those in the ground-truth video while allowing natural variation across frames and providing a broader, more informative reward signal. This similarity serves as the reward signal for our facial reward model, formalized as:

$$s_i = \frac{1}{F} \sum_{j=1}^F \cos(\phi(\hat{x}_i), \phi(x_j)), \tag{5}$$

$$R_{face} = \frac{1}{F'} \sum_{i=1}^{F'} s_i, \tag{6}$$

where $X = \{x_1, \ldots x_F\}$ are ground-truth frames, $\hat{X} = \{\hat{x}_1, \ldots \hat{x}_{F'}\}$ are generated frames, $\phi$ is the face encoder, and $\cos$ denote cosine similarity. $s_i$ is the average similarity between each generated frame $i$ and all ground-truth frames, and $R_{face}$ is the score of the face reward model.

### 4.3 KL-DIVERGENCE REGULARIZATION

We propose a multi-step KL-divergence regularization over the reverse-time sampling trajectory to stabilize training and mitigate reward hacking. Rather than optimizing solely by reward model, we

Table 1: **Quantitative comparisons.** Our method achieves more consistent face similarity than the baseline, without compromising its performance on other dimensions.

| Method | FaceSim↑ | SC↑ | BC↑ | AQ↑ | IQ↑ | TF↑ | DD↑ | MS↑ |
|---|---|---|---|---|---|---|---|---|
| In-house I2V model | 0.4769 | 0.9768 | 0.9777 | 0.6641 | 0.7291 | 0.9895 | 8.93 | 0.9943 |
| *W/ reward model* | **0.6960** | 0.9811 | 0.9810 | 0.6641 | 0.7259 | 0.9913 | 8.31 | 0.9951 |
| Wan 2.2 A14B (Wan et al., 2025) | 0.5780 | 0.9508 | 0.9705 | 0.6588 | 0.7273 | 0.9676 | 19.45 | 0.9804 |
| *W/ reward model* | **0.6942** | 0.9536 | 0.9720 | 0.6607 | 0.7253 | 0.9690 | 19.17 | 0.9813 |

| Input image | In-house I2V | *W/ reward model* | Wan 2.2 | *W/ reward model* |
|---|---|---|---|---|

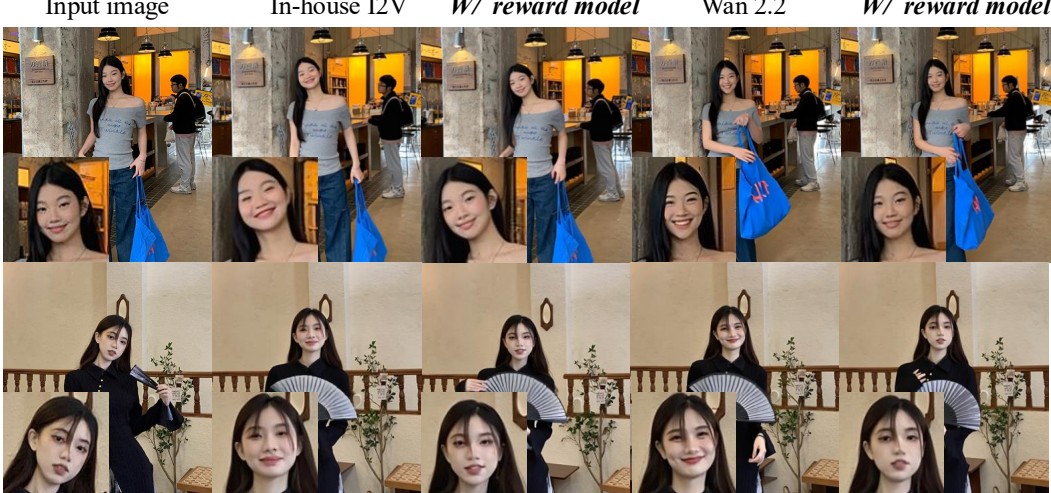

Figure 4: **Qualitative comparison before and after integrating our framework.** Our method achieves more stable generation and superior identity preservation compared to the baseline.

constrain the sampling process from $x_T$ by penalizing deviations between the optimized model $\theta$ and the original model $\theta_{ref}$ at each gradient step:

$$D_{KL}(p_\theta(x_{0:T})||p_{\theta_{ref}}(x_{0:T})) = \sum_{t=1}^{K} \omega_t D_{KL}(p_\theta(x_{t-1}|x_t)||p_{\theta_{ref}}(x_{t-1}|x_t))$$

$$= \sum_{t=1}^{K} \omega_t'||v_\theta(x_t,t) - v_{\theta_{ref}}(x_t,t)||^2. \tag{7}$$

where $p_\theta(x_{0:T})$ is the path distribution induced by the sampling procedure starting from $x_T$ to $x_0$, $\omega_t$ and $\omega_t'$ are step weights, and $v_\theta$, $v_{\theta_{ref}}$ denote the velocity parameterizations. As shown in Fig. 3, the model trained with KL-divergence regularization maintains a low and stable KL divergence, which indicates the regularization updates are well-constrained, preventing large-scale shifts.

## 5 EXPERIMENTS

### 5.1 EXPERIMENTAL SETUP

**Implementation Details.** We utilize both our in-house I2V model (15B dense model with both MMDiT block and single DiT block) and Wan2.2 27B-A14B (Wan et al., 2025) as our image-to-video foundational model. For Wan2.2 27B-A14B (Wan et al., 2025), we keep all modules frozen except the low-noise expert model. In order to improve training efficiency, we employ Wan2.2-Lightning, the distilled version of Wan2.2 (Wan et al., 2025) which requires only 8 steps without the need of Classifier-free guidance (CFG). For the training process, we employ the Adam optimizer configured with a learning rate of 2e-5, and train 100 steps with a batch size of 64. The truncation gradient step $K$ is 4, the facial reward weight is 0.1, and the KL-divergence loss weight is 1.

Figure 5: **Qualitative comparison with other methods.** Our method preserves identity information more faithfully than other methods.

Table 2: **Comparison with other methods.** Our method achieves the highest face similarity among all compared methods.

| Method | FaceSim↑ |
|---|---|
| Wan 2.2 (Wan et al., 2025) | 0.5780 |
| + MoCA[†] (Xie et al., 2025) | 0.5820 |
| + Concat-ID[†] (Zhong et al., 2025) | 0.6056 |
| + DPO[†] Rafailov et al. (2023) | 0.6284 |
| **+ Ours** | **0.6942** |

Table 3: **Ablation study on different training frameworks.** Our method outperforms SFT and CLIP reward in preserving face similarity.

| Method | FaceSim↑ |
|---|---|
| Wan2.2 (Wan et al., 2025) | 0.5780 |
| + SFT | 0.6392 |
| + CLIP | 0.6099 |
| **+ Ours** | **0.6942** |

**Datasets.** We collect high-quality videos with 960p resolution from the Internet and detect faces using SCRFD (Guo et al., 2021). To emphasize small-face scenarios, we retain clips in which the largest face bounding box per frame does not exceed $100 \times 100$ pixels. We discard videos with insufficient face coverage ($< 40\%$ of frames containing a detectable face) and use Qwen2.5-VL (Bai et al., 2025), a vision–language model, to remove videos where faces are occluded by objects (e.g., phones, masks). For the remaining data, we extract face embedding of each face frame in the video.

**Evaluation metrics.** We evaluate the identity consistency on a small-face evaluation set of 600 scenes, by computing face similarity (FaceSim) between the generated frames and the input image. In addition, to comprehensively evaluate the overall quality of generated videos, we also report VBench-I2V (Huang et al., 2024) metrics on its official evaluation set, including Subject Consistency (SC), Background Consistency (BC), Aesthetic Quality (AQ), Imaging Quality (IQ), Time Flickering (TF), Dynamic Degree (DD), and Motion Smoothness (MS).

## 5.2 COMPARISONS

**Comparisons on facial reward.** To valid the effectiveness of our facial reward model, we use the in-house I2V model and Wan2.2 I2V 27B-A14B (Wan et al., 2025) as baselines and compare their performance before and after integrating our framework. As shown in Table 1, the FaceSim metric improves markedly: face similarity for the in-house I2V model increases by $45.9\%$, and Wan2.2 (Wan et al., 2025) improves by $20.1\%$. For general video quality metrics in VBench-I2V (Huang et al., 2024), introducing the reward model does not degrade the models' original capabilities. Fig. 4 further provides qualitative comparisons, where our method exhibits more stable generation and stronger identity preservation.

**Comparisons with other methods.** Lacking I2V identity-preservation baselines, we adapt two state-of-the-art T2V methods, MoCA[†] (Xie et al., 2025) and Concat-ID[†] (Zhong et al., 2025), to I2V model (Wan et al., 2025) and compare them with our method. MoCA[†] (Xie et al., 2025) supervises identity consistency through a latent space identity loss, whereas Concat-ID[†] (Zhong et al., 2025) appends reference face tokens to the video token sequence and employs 3D self-attention for fusion. As shown in Table 2, our method achieves markedly better facial consistency than these two methods, indicating that facial reward-guided learning achieves better alignment with the desired outcomes. This advantage is further validated qualitatively in Fig. 5, where our method exhibits superior identity retention. We further evaluate multi-person scenarios in Appendix A.3, where our method naturally generalizes and demonstrates improved consistency.

Table 4: **Ablation study on reward hacking.** Our method enhances facial consistency without noticeable hacking.

| | Wan2.2 | W/o KL | W/o FSM | Ours |
|---|---|---|---|---|
| FaceSim↑ | 0.5780 | 0.7544 | 0.7388 | 0.6942 |
| Hacking↓ | 7% | 58% | 52% | 10% |

Table 5: **Ablation study on different initial gradient steps.**

| | high noise | low noise |
|---|---|---|
| FaceSim↑ | 0.6456 | **0.6942** |
| Dynamic Degree↑ | 18.98 | **19.17** |

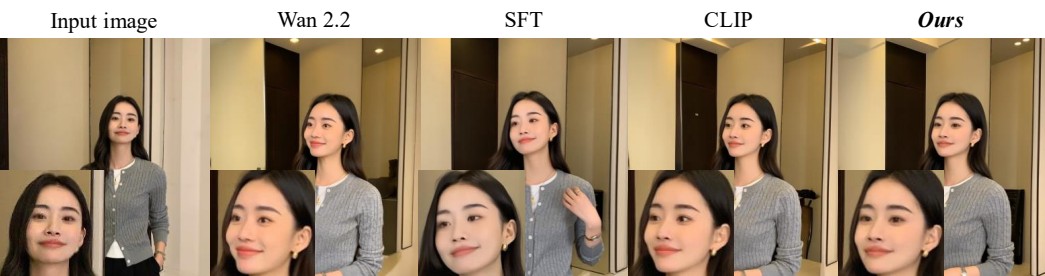

Input image     Wan 2.2     c SFT     CLIP     *Ours*

Figure 6: **Visual comparison on different training frameworks.** Our method achieves more stable generation and superior identity preservation compared to others.

Input image     Wan 2.2     W/o KL     W/o FSM     *Ours*

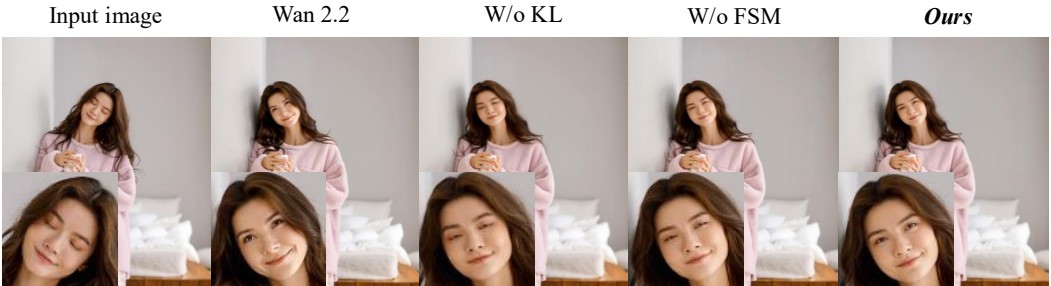

Figure 7: **Ablation study on reward hacking.** Without KL-divergence regularization or the FSM module, the generated video overly adheres to the input image, resulting in facial rigidity and reward hacking phenomenon. However, our method enables accurate, expressive prompt-following behavior, such as opening eyes.

**Comparison with DPO.** By bypassing an explicit reward model, DPO (Rafailov et al., 2023; Wallace et al., 2024) learns directly from human preferences and simplifies the pipeline. Yet the objective is purely relative: gradients come from preferred–rejected likelihood differences, leaving the method vulnerable to confounds and without calibrated, absolute utility. However, reward-guided learning uses pretrained identity encoders to directly provide dense, calibrated signals. Fig. 5 and Table 2 show that our reward-based method outperforms DPO in terms of face similarity.

## 5.3 ABLATION STUDY

**SFT.** To evaluate the effectiveness of our reward-based training framework, we compare it with supervised fine-tuning (SFT). Specifically, we incorporate facial similarity as a loss function into the training objective as an auxiliary loss during supervised fine-tuning. This encourages the model to preserve identity during generation. Despite this explicit supervision on face similarity, we observe that SFT still struggles to preserve face identity. In contrast, our reward-based approach leads to more robust and consistent performance, as shown in Fig. 6 and Table 3.

**Reward model.** We employ the ArcFace model (Deng et al., 2019) as our facial embedding extractor and compute the similarity between embeddings of generated and ground-truth videos. To validate the effectiveness of our reward model, we also experiment with the CLIP image encoder (Radford et al., 2021) to extract facial embeddings and compute similarity scores as the reward signal. However, as shown in Fig. 7 and Table 3, the CLIP-based encoder underperforms compared to Ar-

cFace, indicating that ArcFace is better suited for capturing fine-grained facial identity features in our framework.

**Different choices of gradient steps.** To evaluate the effectiveness of selecting final gradient steps for backpropagation, we conduct experiments comparing high-noise and low-noise initial steps. As shown in Tab. 5, using final (low-noise) gradient steps achieves better identity consistency.

**Reward hacking.** Simply using the face reward model to optimize the model can easily lead to reward hacking, where the generated faces in the video over-adhere to the input image, resulting in stiff faces, lack of expression, and poor response to prompt. To address this, we propose a novel facial scoring mechanism (FSM) and a multi-step KL-divergence regularization to enhance general-ization. As shown in Fig. 7, our method ensures facial consistency while providing better response to prompt. We further leverage the state-of-the-art video understanding VLM Gemini 2.5 Pro (Co-manici et al., 2025) to identify facial reward hacking in generated videos, reporting the hacking rate on the evaluation set as a metric. As shown in Table 4, while ablating the KL regularization and FSM leads to higher facial similarity, it significantly increases the reward hacking phenomenon. More details on the hacking metric are provided in the appendix.

## 5.4 HUMAN EVALUATIONS

We conduct a user study to assess our method from a human perspective. Specifically, we randomly sample 50 images from the small-face evaluation set and we use different approaches to generate videos with identical settings. During the study, participants are shown the input image and two videos (ours vs. baseline) in randomized order. A total of 96 volunteers are invited to select the video that performs better in terms of Identity Preservation, Visual Quality, Text Alignment, and Motion Amplitude, or choose "no preference" if uncertain. The results, summarized in Fig. 8, indicate that our method achieves better identity preservation than baselines, while performing comparably in other dimensions.

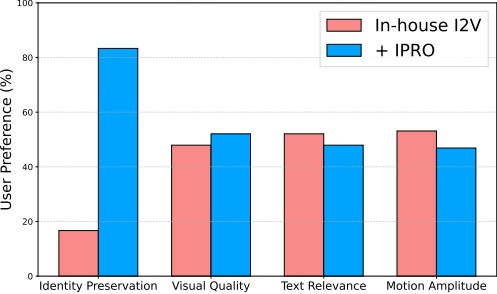 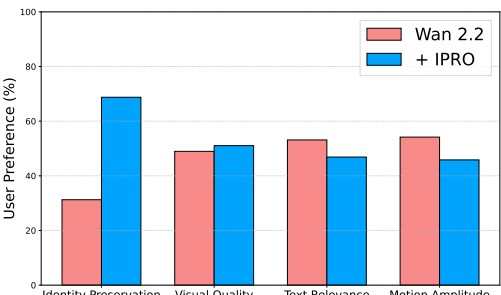

Figure 8: **User study.** User study results show that our IPRO significantly outperforms both the in-house I2V model and Wan 2.2 I2V in identity preservation.

## 6 CONCLUSION

In this paper, we present IPRO, a reinforcement learning–based framework for image-to-video dif-fusion models that substantially improves identity preservation without modifying the underlying architecture or introducing auxiliary identity modules. Our method addresses key shortcomings of prior approaches by (i) leverage a facial reward model to enhance identity consistency (ii) introduc-ing a robust facial scoring mechanism that aggregates multi-view facial features from ground-truth videos, and (iii) stabilizing learning with a multi-step KL regularization that preserves the founda-tion model's original capabilities. We hope that our work can bring identity preservation in I2V and its reward framework into the sight of a broader community and motivate further research.

**Limitations and Future Work.** This work focuses on facial identity preservation in image-to-video generation. However, preserving identity also entails consistency of non-facial attributes (jewelry, accessories, clothing), which remain underexplored. We plan to investigate the unified identity reward model that covers these aspects in future work.

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

# A APPENDIX

## A.1 THE USE OF LARGE LANGUAGE MODELS(LLMS)

We used large language models solely for language polishing (grammar and spelling correction, improving clarity and flow, and shortening long sentences). Specifically, LLMs were not used to conceive ideas, design the method or experiments, implement code, analyze results, write proofs, or conduct literature searches or citation selection. All LLM suggestions were reviewed and substantively edited by the authors; all technical statements, formulas, algorithms, and conclusions are authored and verified by us.

We also used a vision-language model (VLM) as an auxiliary metric to detect "reward hacking" in generated videos—i.e., over-adherence to the first frame that yields unnaturally rigid faces, suppressed expressions, and poor responsiveness to prompts. The vision-language (VLM) was given the following prompt:

"Analyze the following video and evaluate whether it exhibits signs of 'reward hacking' in the context of facial consistency optimization. Specifically, determine if the face remains too rigidly consistent with the first frame throughout the video, resulting in unnatural or overly static facial appearance, lack of natural expression changes, minimal facial dynamics, or absence of expected motion (e.g., subtle shifts due to speech, emotion, or camera movement). Consider the following aspects:

- Facial Motion: Is there realistic and natural variation in facial expressions or muscle movements across frames?
- Consistency vs. Stiffness: While the identity remains consistent, does the face appear unnaturally frozen or overly stabilized?
- Contextual Appropriateness: Given the content (e.g., talking, emotional expression, head movement), is the level of facial motion appropriate?
- Visual Artifacts: Are there any signs of blending, warping, or smoothing artifacts that suggest aggressive enforcement of similarity to the first frame?

Is the video over-optimized (i.e., reward hacking)? Only answer yes or no"

## A.2 FACE SCORE VARIATION WITH TRAINING STEPS

The face score on the training set increases steadily throughout training, indicating that the identity reward effectively enhances facial identity preservation.

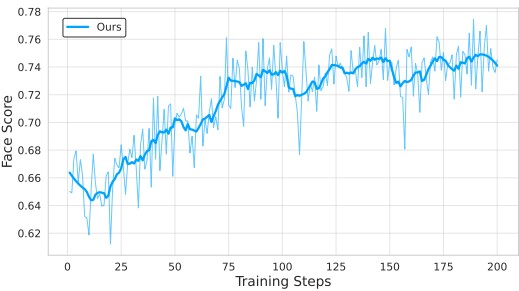

Figure 9: **Face score evolution during training in training set.** Scores increase steadily with training steps, indicating that the face reward effectively improves identity consistency.

## A.3 VISUALIZATIONS OF MULTI-PERSON SCENARIOS

Compared to injecting facial features directly into the base model, our RL-based approach not only requires no architectural modifications or additional modules, but also naturally generalizes to multi-person scenarios. Concat-ID[†] (Zhong et al., 2025) injects reference face tokens to the video token sequence and employs 3D self-attention for feature fusion. However, as the number of characters

in a scene increases, this method struggles to maintain consistent identity representations across frames. In contrast, as shown in Fig. 10, our method achieves superior identity consistency in complex multi-person settings.

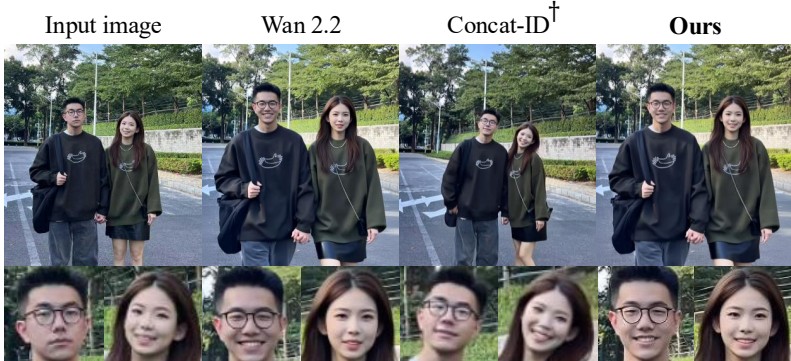

Figure 10: **Visual comparison on multi-person scenes.** Our method preserves identity consistency across frames better than other methods.

### A.4 SOCIAL IMPACTS

Our RL-based image-to-video diffusion method, which improves human identity preservation, offers clear benefits for creative production, accessibility, and scientific research. However, it may also amplify risks of non-consensual deepfakes and misinformation. To mitigate these risks, we will work continuously with legal and ethics experts and impacted communities to iteratively advance safety measures while maintaining the technology's benefits.

