# OpenReview forum: "Identity-Preserving Image-to-Video Generation via Reward-Guided Optimization"
_ICLR.cc/2026/Conference — ICLR 2026 Conference Withdrawn Submission_

### Official Review · Reviewer_SX3p · 2025-10-28

**Soundness:** 4
**Presentation:** 3
**Contribution:** 4
**Rating:** 6
**Confidence:** 5

**Summary:**

To improve the identity consistency of image-to-video models when the reference image contains a low-resolution human face and the motion prompt involves large-scale movements, this work proposes IPRO, a reinforcement learning–based framework. Experiments conducted with state-of-the-art video generation models, particularly Wan 2.2 A14B, demonstrate the effectiveness of IPRO. Interestingly, IPRO enhances identity preservation capabilities during the post-training phase of the base model through reward feedback, without requiring modifications to the model's architecture or the introduction of additional modules.

**Strengths:**

1. IPRO is simple and effective, requires no modification to the base model's architecture, and can serve as a post-training approach to further enhance the base model's performance.

2. This work has significant potential: by modifying the reward model and reward function, IPRO can improve other capabilities of the base video model, not limited to identity consistency.

**Weaknesses:**

1. As the first equation in the Method section, Equation 3 does not include the ground truth, which raises some confusion about whether the ground truth is involved in the reward calculation.

2. The comparative experiments with DPO are impressive; however, many current image/video generation methods (e.g., Flow-grpo [1]) employ GRPO rather than DPO to enhance the performance of the base models. Including additional comparisons with GRPO could further improve the quality and completeness of this work.

[1] Flow-grpo: Training flow matching models via online rl.

**Questions:**

1. There is a typo on line 179: “IPROpredicts” should be corrected to “IPRO predicts”.

If the authors address my concerns, I may consider increasing the score.

---

### Official Review · Reviewer_KFZv · 2025-11-02

**Soundness:** 2
**Presentation:** 2
**Contribution:** 2
**Rating:** 2
**Confidence:** 4

**Summary:**

This paper proposes IPRO, a reinforcement learning–based optimization framework for improving identity preservation in image-to-video (I2V) diffusion models. The method uses a facial reward signal derived from an identity encoder (ArcFace) to fine-tune pretrained diffusion models without altering their architecture.

**Strengths:**

Architecture-agnostic: Does not modify the base model, making it plug-and-play for large diffusion systems.

**Weaknesses:**

1. Novelty: The concept of introducing an additional identity-preserving objective in diffusion-based generative models is not new.
Several prior works, such as DCFace [1], ID3 [2], ID-Booth[3] and many others, have already incorporated identity-related losses or conditioning into diffusion models to preserve subject consistency.
The novelty here primarily lies in the reinforcement-learning formulation rather than the introduction of an identity loss itself, which should be made clearer and positioned more accurately with respect to prior work.

2. Metric–objective mismatch:
The evaluation relies mainly on FaceSim (mean cosine similarity of ArcFace embeddings), which measures static image similarity rather than temporal identity consistency.
A model can achieve high FaceSim simply by producing nearly frozen faces (reward hacking) while still failing to maintain identity under motion or expression change.
3. Need for verification-based metrics:
Identity consistency should ideally be assessed with verification metrics such as EER (Equal Error Rate) or FMR1000, etc, which directly evaluate whether a consistent identity is recognized across frames or sequences.
Using these would better reflect whether the same person is preserved, not just similar embeddings per frame.
Reproducibility:
Evaluation uses proprietary datasets and large in-house models, which limits independent verification.

Evaluation of identity is limited to 1 face recognition model, raising questions about generalizability and validity of the results

[1] Minchul Kim, Feng Liu, Anil K. Jain, Xiaoming Liu:
DCFace: Synthetic Face Generation with Dual Condition Diffusion Model. CVPR 2023: 12715-

[2]	Jianqing Xu, Shen Li, Jiaying Wu, Miao Xiong, Ailin Deng, Jiazhen Ji, Yuge Huang, Guodong Mu, Wenjie Feng, Shouhong Ding, Bryan Hooi:
ID3: Identity-Preserving-yet-Diversified Diffusion Models for Synthetic Face Recognition. NeurIPS 2024
[3] Darian Tomasevic, Fadi Boutros, Chenhao Lin, Naser Damer, Vitomir Struc, Peter Peer:
ID-Booth: Identity-consistent Face Generation with Diffusion Models. CoRR abs/2504.07392 (2025)

**Questions:**

Have you evaluated IPRO using verification-based metrics (e.g., EER, FMR1000 etc) to quantify temporal identity consistency?
What is the conceptual difference between the presented approach and previous ones (ID3 etc.) ?

How does IPRO perform under different face recognition models (ArcFace vs. ElasticFace,  vs. AdaFace)?

How sensitive is the reward signal to noise or occlusions in face detection?

Can IPRO generalize to non-facial identity cues such as clothing or body structure?

---

### Official Review · Reviewer_9q9n · 2025-11-03

**Soundness:** 2
**Presentation:** 2
**Contribution:** 2
**Rating:** 4
**Confidence:** 4

**Summary:**

This paper introduces Identity-Preserving Reward-guided Optimization (IPRO), a reinforcement learning-based framework for improving identity consistency in image-to-video (I2V) diffusion models. Rather than modifying model architectures or introducing additional identity modules, the method optimizes the generation process by backpropagating a facial reward signal, computed via a facial feature embedding model, through the final denoising steps of the diffusion chain. The authors further design a multi-view facial scoring mechanism (FSM) and employ multi-step KL-divergence regularization to stabilize learning and address reward hacking. Results are demonstrated on both public and in-house I2V models, with quantitative and qualitative comparisons to prior methods highlighting improvements in identity preservation.

**Strengths:**

1. The paper targets a salient and underexplored challenge in I2V generation: maintaining identity consistency, particularly for small faces and long video sequences where identity drift is prominent.

2. The RL-based reward-guided tuning framework is methodologically elegant, introducing identity-specific optimization without modifying the model architecture or requiring additional subject modules.

3. Extensive experiments, including ablation studies and human evaluation, offer a comprehensive assessment.

**Weaknesses:**

1. While the reward optimization and KL-regularization are justified intuitively and via prior art, the theoretical properties of the combined objective are underexplored. For example, the interaction between RL-style reward and the original model’s distribution in the presence of the KL penalty is only empirically motivated. There’s no discussion of the convergence characteristics or the effect on the underlying generative process’s distribution, particularly in high-variance or low-signal reward settings.

2.  The approach is heavily dependent on the ArcFace reward model. While ablations demonstrate its empirical superiority, there is little discussion about the limitation of such pretrained facial encoders. Known sensitivities to pose, illumination, and motion blur are not extensively discussed. How does the reward model respond to occlusions or low-res faces, and could it be gamed by artifacts? A more extensive error analysis, perhaps with per-attribute breakdowns, would provide additional confidence.

3. It is difficult to observe that the proposed method can achieve superior identity consistency in generated from Fig.1.

4.  The weighting of the KL-divergence loss, step weights, and truncation steps ($K$) are crucial hyperparameters (Section 5.1), but there is no sensitivity analysis demonstrating robustness to these choices. Given the importance of balancing identity improvement without catastrophic forgetting or reward hacking, more systematic ablation or reporting on these choices is warranted.

**Questions:**

As shown in weaknesses.

---

### Official Review · Reviewer_5wrN · 2025-11-03

**Soundness:** 2
**Presentation:** 2
**Contribution:** 2
**Rating:** 4
**Confidence:** 4

**Summary:**

This paper proposes IPRO, a reinforcement learning-based framework for improving identity preservation in image-to-video generation. The method leverages a facial identity reward model, a novel facial scoring mechanism, and KL-divergence regularization to enhance identity consistency without modifying the base model architecture. Experiments on Wan 2.2 and an in-house I2V model show improvements in face similarity metrics.

**Strengths:**

The problem of identity preservation in I2V generation is important and underexplored.

The proposed facial scoring mechanism and multi-step KL regularization are novel and well-motivated.

The paper provides extensive experiments, including comparisons with adapted T2V methods and ablations.

The method does not require architectural changes, making it more scalable.

**Weaknesses:**

Lack of strong baselines: The comparisons are mainly against adapted T2V methods (e.g., MoCA, Concat-ID) rather than native I2V identity-preservation methods. This weakens the claim of superiority.

Limited generalization: The method is heavily focused on facial identity, while other aspects of identity (e.g., clothing, accessories) are not addressed. This limits the scope and practical impact.

The method lacks novelty.

**Questions:**

No Questions

---

### Note · Authors · 2025-11-14

I have read and agree with the venue's withdrawal policy on behalf of myself and my co-authors.